# Claiming Justice: An Analysis of Child Sexual Abuse Complainants' Justice Goals Reported during Investigative Interviews

Robyn L. Holder *, Dirkje Gerryts, Francisco Garcia and Martine Powell

Griffith Criminology Institute, Griffith University, Brisbane 4122, Australia
* Correspondence: r.holder@griffith.edu.au

**Abstract:** Investigative interviewing of children who report sexual victimisation focuses on helping children tell in their own words what happened. Children may say other things important to them such as their justice goals. We conducted the first research into this possibility in an exploratory analysis of 300 transcripts of actual interviews with child complainants aged 3 to 15 years. Building on an earlier study involving adults, we explored what goals children may articulate, when in the interview process their goals are relayed and in response to which interviewer prompts. Our analysis revealed that most children did articulate one or more justice goals during these interviews, especially their desire for acknowledgement of the victimisation and its wrongfulness. Children articulated their justice goals spontaneously and largely without any direct prompting by the police officer. These findings suggest that there is more that institutions [and researchers] can learn from carefully listening to children and understanding them as agents claiming justice.

**Keywords:** children; sexual victimisation; police interview; justice goals





## 1. Introduction

Supporting children's disclosures of sexual victimisation presumes that adults understand children's motivations to report such information. With disclosure at the centre of attention, much research has explored the barriers and disincentives to children telling authorities what has happened to them (for a review, see Alaggia et al. 2019). However, this focus may have obscured what may motivate children and young people to disclose the abuse. Helping children to say 'what happened' may presume that this is all they may have to say to authorities. What would they like to come about from telling authorities of their experiences? In particular, what aspirations might they have in telling a police officer about the abuse? The present study explored if, in a forensic interview, children expressed motivations for the inherent possibility of justice that the police context embodies. We wondered if, in coming to speak to a police officer about the sexual victimisation, children might begin to articulate outcomes that they envisage criminal justice could provide. That is, that children might be agents in claiming justice.

Reporting to police potentially opens a pathway to attaining goals that criminal justice can provide victims. For adults, the motivations for reporting crime and violence to authorities range from desires for self-protection and the protection of others (Felson et al. 2002; van Dijk et al. 2008) to the social influence of family and friends (Greenberg and Ruback 1992). While motivations for reporting to authorities are not the same as the goals a person may seek, taken together, the research shows that, when seeking justice, adult victims think about objectives for themselves, for the violent person and for their community of others (Holder 2018). This range of adult concerns is reflected to some extent in research with children. In test studies, children have been shown to react to the intentionality and consequence of perceived injustice and are interested in punishing a wrongdoer and in compensating the victim (Marshall et al. 2021; Miller and McCann 1979;

Smith and Warneken 2016). For children, as well as adults, the idea of 'justice' is multi-layered and contextual (Sen 2011). However, much of this research is laboratory-based and uses vignettes that describe injustice or harm that happens to another.

Looking for first-person insights, other studies have analysed interviews or interview transcripts with children disclosing suspected sexual abuse. None use justice as an interpretive frame. Instead, they examined children's motivations to and expectations about their disclosure. A United States (U.S.) study examined children's reasons for telling about sexual abuse, their delays in telling and why they came to tell (Schaeffer et al. 2011). In this research, interviewers asked 191 children a direct question during the forensic interview. Researchers identified stimuli internal to the child, external facilitation to disclose and the child being confronted with direct evidence of the abuse. Similarly interested in disclosure, other researchers analysed 154 interview transcripts (from British and U.S children) to identify any 'expectations' held by the child of the consequences of telling about the abuse (Malloy et al. 2011). The most common consequences identified in the transcripts were possible physical harm to and negative emotions of the child and 'jail/legal consequences for the suspect' (page 8). In a later study of 204 child interview transcripts, these same researchers demonstrated that most children described the person to whom they first disclosed and 38% explained 'why their abuse came to be known to others [but] without mentioning any explicit preference or expectation of belief' associated with the recipient (Malloy et al. 2013, p. 249). In the research we describe in the present article, we build on these studies use of investigative interview transcripts. Using a justice frame, ours is the first to explore the justice goals victimised children may have in disclosing abuse to police interviews.

## 2. Justice Goals

Previous studies with adult victims have described their justice *needs* in relation to their experiences of justice authorities (Herman 2005; Sebba 1996). More recently, studies focused on the public-facing system of criminal justice have come to use the more outward-directed term, justice *goals*.

The justice goals of victims are commonly ascribed to preferences for punishment for intentional wrongdoing (Carlsmith 2006). However, one laboratory study by social psychologists, Gromet and Darley (2009), sought to understand which justice concerns underpinned people's responses to wrongdoing. Their insight was that people have a 'full range of justice concerns' and 'the ultimate goal of achieving justice' is reached by the progressive realization of subgoals (ibid. 2009, pp. 2, 4). Their study used a series of vignettes offered to university student participants in which they were to act as the decision-making judge. In each vignette, the goals were 'punishing the offender, rehabilitating the offender, restoring the victim, reinforcing community values, and restoring the community' (p. 10). The study found that people 'view the satisfaction of multiple justice goals as an appropriate and just response to wrongdoing' (p. 1) and that these are directed at different 'justice targets' (the offender, the victim, and the community) (p. 10).

A longitudinal prospective Australian study drew on these insights to explore the justice goals of adult victims of violence following the police charging of another person with a criminal offence against them (Holder 2018). This study confirmed that, similar to third-party evaluations, direct victims involved in a 'real' process seek multiple goals. At three time points, the study asked victims their motivations for reporting, prospective verdict and sentence preferences, retrospective assessments of decisions reached by authorities and their overall assessment of justice. Victims' goals coalesced in two main areas: the quality of interpersonal treatment and process outcomes as two sides of a single concept of *justice*. Similar to Gromet and Darley, Holder also found that people directed their goals towards themselves as victim, for the offender and for their community of others.

Synthesising the findings of victim-related research, Daly distinguished between measures focused on victim well-being (described as a justice *need*) and measures that could be ascribed to victim engagements with particular justice mechanisms (described

as a justice *interest*) (Daly 2017, pp 114–15). For the latter, she identified five elements: participation, voice, validation, vindication and offender accountability. For the current study, we drew on all these studies about justice goals and victims' justice needs to explore their relevance and application to child victims of sexual victimisation when they engage with police investigators.

### 3. Current Study

The current study asked if children reveal goals for justice when disclosing sexual victimisation to police in an investigative interview, what were these goals and if their goals distributed in any pattern depending on characteristics of the victim, the alleged perpetrator, the incident(s) or other features. Given the ethical and practical challenges conducting research with children, we used transcripts of investigative interviews conducted by trained police as a data source. The interview transcripts possess a number of clear research advantages as the direct, spontaneous and contemporaneous words of the child, albeit words constrained within another's questions. However, the investigative interview is not designed for the articulation of future-oriented goals, such as justice goals. Rather, it is backward-looking on an event or series of events. Primarily, children respond to questions put to them by the investigator. Nonetheless, we hypothesised that children would provide indications, if not direct expressions, of their justice-oriented preferences given the context of the interview. As we explain later in this article, interpretations of the words used by the child in an investigative interview necessarily required grappling with ambiguity, temporal anchoring and associations.

### 4. Method

*4.1. Data Source*

In Australia, specially trained police officers conduct and audio or visually record an interview with children about allegations of sexual offences against the child. If the investigating officers have reasonable suspicion that an offence has occurred, then a brief of evidence is provided to the public prosecutor for a separate assessment if there is sufficient evidence to proceed to trial. The brief will usually include a transcription of this interview. For this study, 600 hard-copy transcripts from three different police services in Australia held within Griffith University's Centre for Investigative Interviewing archive were selected for analysis. The selection criteria of transcripts were that the person being interviewed was aged 18 and under and the child was interviewed about possible sexual abuse. After analysing half of the transcripts, data saturation was reached and further coding ceased (Saldaña 2021). The final sample of 300 transcripts comprised interviews conducted between 2004 and 2014. The researchers had no other information on the victim, the incident, the outcome of the interview or the interviewer. Full ethical approval was received from Griffith University Human Research Ethics Committee (GU Ref No: 2018/512). All transcripts were deidentified prior to use, and the researchers had no access to any supplementary case information.

*4.2. Procedure*

4.2.1. Developing Coding Instrument

We adapted coding instruments from previous justice research (Daly et al. 2019). We defined justice goals as the 'child's comments or implied views about their individual justice goals. These may be expressed as aspirations for what they want from the process of reporting to police or as their motivations for reporting [framed by researchers as sub-goals to ultimate goal of achieving justice]' (emphasis in original). Although terms used to embody justice needs and goals evolved through the piloting of the instruments, we began with Daly's five elements of participation, voice, validation, vindication and offender accountability.

A draft coding instrument was applied by the first two authors to five sample interview transcripts. This initial screening exercise revealed that the transcripts also contained

sufficient information to code for personal characteristics of the child, particulars of the victimisation and the offender(s) plus aspects of the interview itself. However, it was apparent that the initial wording of justice goals was too broad to apply to the specificity of the child's words in the transcript. A second iteration of the coding dictionary provided definitions for each variable using simplified language and examples from the transcript text. An excel datasheet was developed to document both the code and those words of the child's that demonstrated the coding decision. For example, if coders interpreted a child's words as articulating a justice goal about "wrongfulness" of their victimisation, then the text received a code of 1 = yes, articulated. The relevant transcript text—for example, "*Because I knew it wasn't right*"—and its page reference would be added alongside the code in the Excel datasheet.

We also wanted to learn if children's articulation of their justice goals and needs were prompted in some manner by the interviewer. In investigative interviewing literature, distinctions are made between open-ended questions, specific questions, specific cued-recall questions and minimal encouragers (amongst others) that are used by interviewers (Powell and Snow 2007). However, in the current study, the interviewer prompt was simply described as a direct prompt, indirect prompt or a mix of direct and indirect prompt. A 'direct' interviewer prompt was defined as a 'specific cue made by the interviewer in the transcript that retrieved a specific answer in relation to the child's goals for justice'. For example, a question such as "*Why did you tell your mum*?" was coded as a direct prompt. An 'indirect' interviewer prompt was defined as an interviewer communication in the transcript that was not formulated with the intention of seeking information from the child in relation to their goals for justice. For example, "*mmhmm*" was coded as an indirect prompt. A 'mixed' interviewer prompt was defined as 'a combination of direct and indirect interviewer prompts to elicit a child's goals for justice'. A mixed interviewer prompt was coded when the interviewer asked multiple questions within one statement. For example, "*What was that stuff about? Why did it feel uncomfortable?*" Mixed interviewer prompts were also coded when the interviewer's question was phrased with a indirect prompt first, followed by a direct question. For example, "*okay*" "*what did you tell your mum?*" The moment in the interview (in its structure) where the prompt was conveyed was also included in the pilot coding sheet.

### 4.2.2. Pilot Study

The revised coding dictionary contained 50 variables, of which 21 were demographic and incident variables, and 29 were conceptual variables (justice needs and justice goals). Next, the revised coding sheet was piloted with a further twenty interview transcripts. These transcripts were purposively selected from the main data source: five transcripts per age band 0–5, 6–10, 11–15, 16+ years. Equal proportions were selected from the jurisdictional samples. Each transcript was selected to assess the age of the child (visible on the first page of the transcript). Those transcripts which had an undesirable age range were placed back into the data pool, and the next transcript was selected. Three coders then independently applied the revised coding sheet and coding dictionary to the same twenty transcripts.

### 4.2.3. Inter Coder Reliability

On completion of the pilot, the three coders met to share and discuss coding decisions. Demographic and incident variables were easily located in the transcript and agreed between coders, but the interpretation of conceptual variables was more challenging. We applied a simple percent agreement as the index of inter-coder reliability. We measured the proportion of coding decisions that reached agreement out of all coding decisions made by two or more coders. Percent agreement on the conceptual variables spread across a wide range due to the nature of the coding set. If a coder differed in coding a justice need or justice goal, then the three variables that immediately followed (police interview formats) would invariably be coded in keeping with the primary code decision. Other differences related

to coding decisions between 'unsaid' and 'unclear'. Where differences were identified, the coder referred to the relevant transcript text and discussion amongst coders ensued. Ultimately, coders agreed that, if there was absent information then the correct code was 'unsaid'. If there was relevant text for the need or goal, but it was ambiguous then the code applied was 'unclear'. Over time, the coders became more confident in interpretation and, in regular meetings, clarified with each other any uncertainty. As a result, percent agreement was generally low for the first few transcripts in the pilot and became higher as coding progressed. The percent agreement across all 50 variables ranged between 60% and 92%. Percent agreement for the demographic and incident variables ranged between 60% and 90%. Percent agreement on the conceptual variables ranged between 55% and 100%.

### 4.2.4. The Final Instrument

Following the pilot, the coding sheet and data dictionary were further simplified until final versions were agreed upon for a complete analysis of all transcripts. We coded six justice *goals* identified as aspirations from the child for acknowledgement, perpetrator accountability, for the wrongfulness of the victimisation to be found, for protection/safety and for outcomes of the punishment or rehabilitation of the perpetrator. The goal of acknowledgement was further broken into five components that we came to view as aspects of the child's justice *needs*: to be acknowledged as a victim, to have the harm acknowledged, to be acknowledged they are believed and to be acknowledged they are not blamed. The goal for protection/safety was also broken into four components: protection/safety for themselves [the child], for self and others, for others and for the perpetrator. Table 1 sets out the justice goals and definitions used in the study.

**Table 1.** Study justice goals and definitions used.

| Justice goal | **Definition** **Child's comments or implied views about <u>their own</u> justice goals. These may be expressed as aspirations for what they want from the process of reporting to police or as their motivations for reporting [framed by researchers as sub-goals to ultimate goals of achieving justice].** |
| --- | --- |
| **Acknowledgement** | |
| Acknowledgement as a *victim* | **Acknowledgement** reflects the child's desire to be recognized (in a particular way) by others who are important to the child. <br><br> The child says or implies that they want to be acknowledged that they *are a victim* of the victimization. <br> e.g., *"I was crying"*–when telling mum |
| Acknowledgement of the *harm* | **Acknowledgement** reflects the child's desire to be recognized (in a particular way) by others who are important to the child. <br> The child says or implies that they want acknowledgment that they were **harmed** because of the victimization (hurt, injury or loss). <br> e.g., *"It hurt me"* |
| Acknowledgement as *believed* | **Acknowledgement** reflects the child's desire to be recognized (in a particular way) by others who are important to the child. <br> The child says or implies that they want acknowledgment that they were **believed** about the victimization, that s/he was victimized. They want others to agree with his/her version of the victimization and its impact. <br> e.g., *"He DID do that"* (emphasis in transcript) |
| Acknowledgement they are *not blamed* | **Acknowledgement** reflects the child's desire to be recognized (in a particular way) by others who are important to the child. <br><br> The child says or implies that they want acknowledgment that they are not **blamed** about the victimization or the actions of the perpetrator. <br> e.g., *"It was his fault"* |

**Table 1.** *Cont.*

| | |
|---|---|
| **Perpetrator Accountability** | A desire that alleged perpetrators are called to account and held to account for their actions (taking responsibility)<br><br>Does the child say or imply that they want the perpetrator to take responsibility for their actions (e.g., apology) or be held responsible for their actions (e.g., be convicted)?<br><br>*Example:*<br>*"I told mummy"*<br>*"[It] needs to get solved"* |
| **Wrongfulness** | The child says or implies an aspiration that the act or perpetrators' actions are **found to be wrong**.<br><br>(a)   The child says or implies that the act IS wrong (morally and legally) (Vindication of the law).<br>(b)   The child says or implies that the perpetrator's actions against the child ARE wrong (Vindication of the child).<br><br>*Example:*<br>*"The bad thing"*<br>*"It was wrong"*<br>*"He still shouldn't have done what he did to me"*<br>*"He was rude to us"* |
| **Protection/Safety** | The child says or implies that they are seeking safety/protection for themselves and/or others as reasons for disclosing the victimization.<br><br>*Self*—protecting self. The child says or implies that they did not want the victimization to happen to them again, *e.g., "I said stop"*.<br>*Other*—protecting others. The child says or implies that they did not want the offence to happen to someone else (family, friends), *e.g., "(suspect) is hurting (sister)"*.<br>*Self and Other*—The child says or implies that they want protection for themselves and other individuals.<br>*Perpetrator*–protecting the perpetrator |
| **Punishment** | Does the child say or imply that they want the perpetrator to be punished for their actions? Does the child want the perpetrator to receive a penalty as retribution for the offence?<br>*Example:*<br>*"I don't want him to come home"*<br>*"He is going to get in trouble"* |
| **Rehabilitation** | Does the child say or imply that they want the perpetrator to be rehabilitated to receive treatment to stop the victimization?<br>*Example:*<br>*"If he got help"* |

Phrasing from the child of a justice goal received either a code of 1 = yes, articulated [in the transcript text], 2 = no, not articulated, 3 = goal unsaid or 4 = goal is unclear to the coder. All codes were cautiously applied as the interview transcripts were the only data source. We found some overlap in our interpretations of text demonstrating a justice goal. In these situations, if one piece of text could be interpreted in two ways, then we coded it twice. For example, "*It was his fault*" was coded as a justice need for acknowledgment not to be blamed, as well as a justice goal for wrongfulness to be found.

4.2.5. Interviewer Prompts and Interview Structure

Following an articulation of a justice goal, researchers next coded for interviewer prompts. When relevant phrasing was identified, coders then assessed if 1 = there was an interviewer prompt, 2 = no interviewer prompt, 3 = unsaid in the transcript, 4 = unclear to the coder and 5 = not applicable. The not-applicable code was allocated where there was an interviewer prompt but no justice goal was identified directly following. Additionally, the

nature of the interviewer prompt was coded as 1 = direct interviewer prompt, 2 = indirect interviewer prompt or 3 = a mixture of both direct and indirect interviewer prompts.

Finally, the interview structure was coded in two parts. Section A was the first part of the interview. Usually this included the substantive phase of the interview when an open narrative from the child was encouraged by the interviewer. Section B was the end part of the interview and included a more specific and focused style of questioning by the interviewer. In transcripts where there was no structure or no difference in questioning, we included a code for 'other'. For each identified justice goal and each identified interviewer prompt, then their location in the interview structure was also coded. Appendix A provides definitions for where in the interview structure the interviewer prompts were identified and the nature of the interviewer prompt. Appendix B provides examples of justice goals in the child's words and the associated interviewer prompt.

## 5. Characteristics of the Transcript Sample, Their Victimisation and Disclosure Trajectories

### 5.1. The Sample

The 300 transcripts were interviews of 60 boys and 240 girls aged from 3 to 15 years old (*M* age years = 10.29, *SD* = 3.06). Most children were aged between 8 and 11 years old (48%, *n* = 145); 80 children were aged 12 to 18 years (27%); 63 children were aged 3 to 7 years (21%), and two children were aged 3 years (1%). In ten of the transcripts, we could not identify the age of the child.

The child interviewees described the perpetrators of their victimisation as almost always male (99%, *n* = 297), with two describing female perpetrators. One transcript referred to a case that had both male and female perpetrators. In 280 (93%) transcripts, the child described a single perpetrator. Most perpetrators were described by the children as adults over the age of 18 years (93%, *n* = 280) with nine under the age of 18 years (3%). In 11 transcripts (4%), the age of the perpetrator could not be identified.

Over half of perpetrators (55%, *n* =165) were unrelated to the child. In this category, however, a majority (85%, *n* = 141) were familiar as a neighbour, family friend, friend or boyfriend of the child. A smaller proportion (8%, *n* = 24) were strangers, that is, a person that the child did not know and had not previously met. Related perpetrators were immediate family members (sibling, father, stepfather, mother's boyfriend) (27%, *n* = 81) and other relatives (uncle, grandfather, cousin of the child) (18%, *n* = 54).

### 5.2. Nature of the Victimisation

In most transcripts (59%, *n* = 176), children disclosed multiple victimisation acts. Furthermore, in half of the transcripts (53%, *n* = 159), children described a non-penetrative sexual act. Equal proportions of the transcripts revealed the child describing a penetrative act (23%, *n* = 70) and both penetrative and non-penetrative sexual acts (24%, *n* = 71). The timespan during which the victimisation occurred with the primary perpetrator was analysed as a *single time* (the victimisation occurred within a timeframe of 24 h) or *ongoing* (the victimisation occurred longer than a timeframe of 24 h that could be days, weeks or years). Multiple victimisation acts could occur in either of these timespans. The child described the victimisation as ongoing in 158 of the transcripts (53%) and as occurring a single time in 141 transcripts (47%). In one transcript, the child did not specify the timespan.

Of the 300 child interview transcripts, under a third (28%, *n* = 82) disclosed some communication between themselves and the perpetrator *after* the initiation of victimisation where this communication was connected in some manner to the act/s but was not a description of the acts. For example, "*You are my secret girlfriend, don't tell anyone*". Mostly (71%, *n* = 214), however, communication between the child and the perpetrator was *unsaid*.

We coded comments made by the child that revealed if their attitudes towards or perceptions of the act at the time of victimisation were positive, negative or ambivalent and whether this situational assessment changed after their disclosure to someone other than police and how. A comment interpreted by researchers as *positive* (positively orientated

to the victimisation) could be, for example, *"I liked the attention he gave me"*. A comment interpreted as *negative* (negatively orientated to the victimisation) could be, for example, *"I felt scared"*. A comment interpreted as *ambivalent* (the child might display a mixture of positive, negative and uncertainty) could be, for example, they might not be sure if they liked the attention but did not like the perpetrator's actions. In most transcripts (60%, *n* = 181), children mentioned a negative situational assessment at the time of the victimisation; seven transcripts mentioned a positive situational assessment, and nine children were ambivalent. Just over a third (34%, *n* = 103) did not mention (*unsaid*) a situational assessment made by the child.

*5.3. Disclosure Trajectories*

In the course of their interview, most children (75%, *n* = 225) described having told someone prior to telling the police about their victimisation. Nine children indicated that they did not tell anyone about their victimisation before the police interview, while 66 (22%) of transcripts did not reveal if the child told someone before telling the police (that is, prior police disclosure was *unsaid*). Of those children who told someone about the sexual abuse before the police interview, most mentioned a single person (70%, *n* = 157), and 69 (31%) children mentioned telling more than one person.

Most of the children who told someone other than police (86%, *n* = 193) used some words that indicated the disclosure was intentional; meaning, the child disclosed the abuse with the aim of revealing the victimisation. For example, *"I told someone, it's not just something that's just come up"*.

Of those who told someone prior to police, 165 (73%) of these disclosures were to a family member. Mostly, this family disclosure was to parents (71%, *n* = 117), particularly their mother (78%, *n* = 91). Children also disclosed the victimisation to siblings (15%, *n* = 25), followed by other family members such as grandparents, aunties or uncles and cousins (14%, *n* = 23). Some children (16%, *n* = 36) told a friend; eight children told a teacher; six children told a counsellor, and ten told some other person.

Most of the transcripts (57%, *n* = 128) did not reveal comments from the child about the nature of the response from the person they initially disclosed the victimisation to. Of the 97 transcripts wherein children did say something about the response from the person they told, their words described that response as affirming the wrongfulness of the victimisation (73%, *n* = 71). For example, *"this isn't your fault"*. Three children used words to describe that they felt blamed for the victimisation after the initial disclosure to a person other than police. For example, *"She [mum] was a bit angry with me"*. Some responses (24%, *n* = 23) could not be coded as *affirming wrongfulness* or *blaming* responses and were coded as *other* response instead. For example, *"she didn't know what to say"*.

**6. Results**

In this section, we answer our three research questions. (1) Did children reveal goals for justice when disclosing sexual victimisation to police in an investigative interview; (2) what were these goals; and (3) were children's justice goals distributed in any pattern depending on characteristics of the victim, the alleged perpetrator, the incident(s) or other features?

RQ 1: Did children reveal goals for justice?

The first part of our analysis required us to examine whether children articulated any goals for justice, and if they did, how many types of justice goals were articulated within each transcript. First, from all 300 transcripts, we counted the number of any justice goal expressed by the child during the interview. In the majority of the transcripts (81%, *n* = 243), children revealed one or more justice goal. Second, of those children who expressed justice goals, we counted the number of goals articulated by the child as identified in the transcript (Table 2). Nearly two-thirds of children (73%, *n* = 177) articulated two or more justice goals.

**Table 2.** Justice goals (n and %) revealed per child/transcript (*n* = 243).

| Number of Justice Goals | Frequency |
|---|---|
| One justice goal | 66 (27%) |
| Two justice goals | 89 (37%) |
| Three justice goals | 56 (23%) |
| Four justice goals | 28 (11%) |
| Five justice goals | 4 (2%) |
| Total child/transcripts | 243 (100%) |

RQ 2: What justice goals did children reveal, when and how in the interview?

Next, we wanted to determine which of the six justice goals was most commonly expressed by children. Table 3 shows that, of the children who articulated a justice goal (*n* = 243), acknowledgment (82%) was the most frequently revealed, followed by an aspiration that the perpetrator's act or actions would be found to be wrong (64%), and a third sought the justice goal of protection or safety (41%).

**Table 3.** Justice goals frequency and sample of children's words (*n* = 243 transcripts).

| Justice Goal | Frequency | Number | Example |
|---|---|---|---|
| **Acknowledgment** | **82%** | **199** | |
| Victim | | 138 | "I knew I needed to tell someone because I was really scared, and I needed somebody to talk to and let it all out" |
| Harm | | 96 | "because I got hurt by that bully" |
| Believed | | 45 | "I didn't think anyone would believe me" |
| Not blamed | | 53 | "I kept it a secret because I didn't want to get in trouble" |
| **Wrongfulness** | **64%** | 155 | "It's not right" |
| **Protection** | **41%** | **99** | |
| Self | | 71 | "I didn't want it to happen again" |
| Other | | 5 | "I don't want it to happen to anyone else" |
| Self and other | | 19 | "other girls said that they were touched inappropriately but they were scared to tell the teacher" |
| Perpetrator | | 4 | "I was trying to protect (suspect)" |
| **Perpetrator Accountability** | **33%** | **81** | "I told him again and again and again to stop but he didn't—that's why I came here" |
| **Punishment** | **14%** | **35** | "so they can hurt that man the way he hurt me" |
| **Rehabilitation** | **1%** | **3** | "so they can help him to stop so he can't do it anymore" |

The frequency of justice goal revealed does not add to 100% because the number of goals mentioned in each of the 243 transcripts varied.

The goals of acknowledgement and protection are further broken down. Table 3 shows that mostly children articulated a desire to be acknowledged as a victim, followed by a desire to have the harm acknowledged. Smaller proportions of transcripts mentioned acknowledgement as a desire to be believed and to not be blamed. For the justice goal of protection (*n* = 99), nearly three quarters indicated this was protection for the self, followed by protection of self and others. Of those four transcripts that revealed a justice goal of protection for the offender, the child was usually an adolescent who disclosed acts committed by a person they described as an older boyfriend.

We found that children mostly revealed their aspirations for justice in the first part of the interview (categorised as section A) (Table 4). Mostly the children articulated the justice goal in response to an indirect interviewer prompt.

**Table 4.** Justice goals, section of interview and nature of prompt ($n$ = 243 transcripts, number times revealed).

| Justice Goals | Total | Interview Section | | | Interview Prompt | | |
|---|---|---|---|---|---|---|---|
| | | Section A | Section B | Other | Direct | Indirect | Mix |
| **Acknowledgment** | | | | | | | |
| Victim | 138 | 116 | 17 | 5 | 16 | 118 | 3 |
| Harm | 96 | 88 | 3 | 5 | 24 | 64 | 8 |
| Believed | 45 | 25 | 16 | 4 | 12 | 32 | 1 |
| Not blamed | 53 | 39 | 12 | 2 | 15 | 37 | 1 |
| **Wrongfulness** | 155 | 136 | 11 | 8 | 14 | 133 | 7 |
| **Protection** | 99 | 74 | 13 | 12 | 21 | 67 | 10 |
| **Perpetrator Accountability** | 81 | 66 | 11 | 4 | 11 | 66 | 3 |
| **Punishment** | 35 | 23 | 11 | 1 | 5 | 26 | 2 |
| **Rehabilitation** | 3 | 3 | 0 | 0 | 0 | 2 | 1 |

Number of times when in the interview the justice goal was revealed and following what interviewer prompt does not add to 100% because the number of goals mentioned in each of the 243 transcripts varied.

RQ 3: Were there any relationships between articulating justice goals and other descriptive variables?

Chi-squared analysis was used to identify if the presence of any children's justice goals were associated with characteristics of the child, the alleged perpetrator, the incident(s) or other features. c. We found significant associations between the articulation of a justice goal and the child's age, victimisation timeframe, multiple sexual abuse acts, type of sexual abuse, post-abuse communication with the perpetrator, disclosure to non-police and the child's assessment of the victimisation.

Age of child

The results showed a significant association between the children's age in the sample and those children articulating justice goals, $\chi^2$ (3, $n$ = 290) = 10.65, $p < 0.014$, phi = 0.19. Irrespective of age group, children articulated goals for justice; more children aged over 8 years did so.

Nature of victimisation

We found a significant association between the types of sexual acts the perpetrator committed and the child's articulation of justice goals $\chi^2$ (2, $N$ = 300) = 16.59, <0.001, phi = 0.23. Types of sexual acts were either coded as acts involving penetration, non-penetration acts or both penetration and non-penetration. Of the children who experienced penetrative act(s), 91% articulated a goal for justice compared to 72% of children who experienced a non-penetrative act and who articulated a goal for justice. We also found that 88% of children who experienced multiple acts of sexual victimisation articulated goals for justice compared to 73% of children who experienced a single act and who articulated a goal for justice. The timespan for victimisation was coded either as taking place in a single time or as ongoing (that is, taking place in periods longer than a 24 h period). We found an association between the timespan of victimisation and justice goals, $\chi^2$ (2, $N$ = 300) = 9.16, $p < 0.010$, phi = 0.17. Of the children who experienced ongoing victimisation, 87% articulate justice goals compared to children who experienced victimisation in a single time (74%) and who articulated a goal for justice.

Child's Communication with Perpetrator

We coded if the child mentioned in the interview that there had been communication between themselves and the alleged perpetrator that took place after the initiation of victimisation and that the communication was connected in some manner to the act/s but was not a description of the acts. We found an association between communication with the perpetrator after the victimisation and children articulating justice goals $\chi^2$ (2,

$N = 300$) = 18.88, <0.001, phi = 0.25. Of the children who disclosed communication between themselves and the perpetrator after the victimisation, 96% articulated a goal for justice.

Prior Disclosure to Non-Police

If the child disclosed that they told someone about the victimisation other than the police, it was coded as a non-police prior disclosure. A significant association was found between non-police disclosure and the child's expression of justice goals $\chi^2$ (2, $N = 300$) = 16.63, <0.001, phi = 0.23. Of the children who told someone before telling the police, 86% articulated a goal for justice. Further, an association was also found between the number of people the child disclosed to before disclosing to police and justice goals $\chi^2$ (2, $N = 300$) = 14.37, <0.001, phi = 0.21. Children who told a person (regardless of the number of people they told) about the victimisation prior to telling police articulated goals for justice (86%, $n = 194$).

Of those who indicated they had told someone else about the victimisation prior to telling police, most did so intentionally. Intentional disclosure was also associated with articulating justice goals $\chi^2$ (3, $N = 300$) = 33.14, <0.001, phi = 0.33. Children who intentionally disclosed to a person prior to telling police mostly articulated justice goals (89%).

Lastly, where children had told someone else prior to telling police, the nature of the response they received was also associated with the articulation of justice goals $\chi^2$ (4, $N = 300$) = 19.98, <0.001, phi = 0.25. Interestingly, both children who received a blaming response articulated goals for justice (100%), as well as children who were validated in the wrongfulness of the victimisation (90%).

Child's Assessment of the Victimisation

The child's assessment of the victimisation at the time of offence was coded as either positive, negative, ambivalent or is unsaid. Chi-squared analysis showed a significant association between the child's assessment of the victimisation at the time of offence and justice goals $\chi^2$ (3, $N = 300$) = 13.47, $p < 0.004$, phi = 0.21. Both children who expressed ambivalence at the time of victimisation articulated goals for justice (100%), as well as those children who expressed negative views about the victimisation (86%).

We also coded if this assessment made of the victimisation by the child changed from the time of victimisation to sometime after victimisation. There was a significant association with justice goals $\chi^2$ (2, $N = 300$) = 8.82, $p < 0.012$, phi = 0.17. Children who changed their assessment of the victimisation were likely to articulate goals for justice (100%) compared to those who did not change their assessment of the victimisation (93%).

## 7. Discussion

In disclosing sexual victimisation to a police interviewer, children also reveal their justice goals. While the police interview is a critical moment for the child to offer an elaborate and accurate account of what happened (Powell and Snow 2007), adult attention is focused on what they need for adult ends. Our study reveals that children say more than what happened, more than the disclosures that adults are listening for. Rather than 'expectations' or 'consequences' (Malloy et al. 2011), justice goals are, in part, what children want to see happen as a result of their disclosure of sexual victimisation. While it is a limitation of the study that investigative interview transcripts are highly scripted questions and answers, our finding aligns with research showing children have a 'desire to see good actions rewarded and bad actions punished' (Bloom 2013, p. 3). Further, a majority of children who articulated a justice goal in response to the 'bad actions' they experienced mentioned two or more goals. For the children in this study, their predominant goals were for acknowledgement, that the perpetrator's actions are found to be wrong and for the protection of themselves and others.

In studies with adult victims, the justice goals identified are similar to those found herein with children. Often referred to as justice *needs* of adult victims (Koss 2010; Sebba 1996), the present study found justice goals similar to research with adult survivors of

sexual victimisation (Herman 2005; Jülich 2006). While those studies shared terms used in our coding, they also commonly used the terms validation and vindication. In our study, children's goals for acknowledgement overlapped with both these: that the child be validated as a victim, that the harm they have experienced is validated and that they be believed and not blamed. We found the term vindication overlapped with children's assessment that perpetrators actions were wrong and should be found to be wrong. Future research could engage more deliberately with these theoretical frameworks to first person (victim) views, experiences and evaluations of justice and critically consider the relevance and usefulness of associated terminology to children as subjects. While our study used deductive coding for theoretically informed justice motivations, future researchers may use inductive coding of transcripts or other qualitative data to explore the aspirations for justice held by child victims of sexual abuse.

Irrespective of age, most children in our study did articulate justice goals. The articulation of justice goals is also associated with elements of the self-assessed seriousness of the victimisation: elements such as it being a penetrative act, multiple acts and ongoing victimisation. Two other associations are important to mention. First, there were associations between the articulation of justice goals, the child's situational assessment of the victimisation and if there was a change to that assessment as revealed in the interview transcript. Second, there was a significant association between the articulation of justice goals and the child having told someone else about the abuse prior to telling police and if there was communication with the perpetrator after the victimisation.

These findings go to the social contexts of children's disclosures. Children's initial primarily negative situational assessment of the abuse and any change away from positive assessments could happen for many reasons. We posit that one could be that the communication between the perpetrator and child or communication between the child and persons whom the child told about the abuse prior to telling police not only affirms the child's discomfort or feeling of being wronged but also help move them to think about what should next happen. As has been identified in studies with adult victims, social support for the victim comprises sharing recognition of the harm, but it also acts to shape the victim's views that what happened is wrong and that this wrong should be disclosed to authorities (Holder 2018; Jensen et al. 2005; Vidmar and Miller 1980). Identifying what is a wrong is as much a social process as is fashioning a just response.

Children's justice goals were also not free-floating but were linked to 'targets' (Gromet and Darley 2009, p. 2). These targets were for themselves (e.g., "*it hurt me*"), their community of others (e.g., for a sister: "*I didn't want it to happen to her*") and also the offender (e.g., "*he done something wrong*"). The target trilogy calibrates with much legal and political philosophy that says *justice* is a project for a community of citizens (Duff 2011), a community that we argue include children. Justice is not an adult-only project. That *justice* does not have a single focus also accords with empirical research into the layered multiple interests in justice that adult victims of sexual victimisation have (Herman 2005; Koss 2010; Jülich 2006).

Most children revealed their justice goals during the free narrative part of the interview and were not directly prompted by the police interviewer to do so. This is also an important finding as the police interview is a controlled and 'high-structure setting' (Castro 2017, p. 149). Nonetheless, the children being interviewed by police not only found space to articulate what concerned them, they were also able to say something early of what they sought as a result of their disclosure. Perhaps, after receiving an elaborate disclosure from the child, interviewers can add future-oriented questions for the child such as: "*What would you like to see happen now?*" Crucially, a single question will likely get only a single answer. As our study showed, children have more than one thing that they want to see happen. Follow-on questions could therefore be: "*And after that? And after that?*" Future research could examine the implications of this approach.

The agentic child identified in our research holds an important place in national and international policy frameworks. However, researchers and policymakers alike struggle to

make or allow this to happen, especially in situations that engage a child's direct interests (Bell 2008). The verbatim transcripts of the child's police interview provide one way to overcome this barrier and to listen again to the child. Other studies using this source have identified similar words and phrases used by the child being interviewed but analysed these in relation to overriding focus on the issue of disclosure (Malloy et al. 2011, 2013). Our study positioned the child as an agent within the public-facing institutions of criminal justice. In doing so, the child emerged as a claimant of justice for themselves as well as for others.

**Author Contributions:** Conceptualization, R.L.H. and D.G.; methodology, R.L.H., D.G., F.G. and M.P.; software, D.G.; validation, R.L.H., D.G. and F.G.; formal analysis, R.L.H. and D.G.; investigation, R.L.H. and D.G.; resources, R.L.H., D.G., F.G. and M.P.; data curation, R.L.H. and D.G.; writing—original draft preparation, R.L.H. and D.G.; writing—review and editing, R.L.H., D.G., F.G. and M.P.; visualization, R.L.H. and D.G.; supervision, R.L.H. and M.P.; project administration, R.L.H.; funding acquisition, R.L.H. and M.P. All authors have read and agreed to the published version of the manuscript.

**Funding:** This research was funded by a Strategic Development Grant from the Griffith Criminology Institute, July 2018.

**Institutional Review Board Statement:** The study was conducted in accordance with the Declaration of Helsinki, and received full ethical approval from Griffith University Human Research Ethics Committee (GU Ref No: 2018/512).

**Informed Consent Statement:** Not applicable.

**Data Availability Statement:** Contact the first named author for data definitions, coding and raw results.

**Conflicts of Interest:** The authors declare no conflict of interest.

### Appendix A. Definitions for Interview Analysis

| Interviewer Set | Definitions |
|---|---|
| Police interview prompt | Was there a verbal expression (e.g., question, gap in question, other technique) by the interviewer that facilitated the child being able to articulate her/his assessment of **wrongfulness?** |
| Where in interview articulate need | Where in the interview was a police prompt used prior to a child articulating her/his assessment of **wrongfulness?** <br> *Section A (the story)* = Substantive phase. The part of the interview where an open narrative is encouraged. <br> *Section B (more probing)* = Further questioning of what happened. The part of the interview involving further specific questioning from interviewer. <br> *Other*—In a case where there is *no structure* or difference in questioning in the interview |
| Nature of prompt | The characteristic of the interviewer's prompt (e.g., question, gap in question, other technique) prior to articulating her/his assessment of **wrongfulness** <br> *Direct*—a specific cue that aims to retrieve a specific answer in relation to the child's justice needs, e.g., "Why did you tell the policeman?" <br> *Indirect*—a cue that provides an opening for some information from the child. The cue is not directed at child's justice goals, e.g., "hmm hmm" <br> *Mixture*—if the interviewer used a combination of direct and indirect prompts to elicit the child's justice needs |

Source: Holder, Gerryts and Powell (2019) Children's Justice Goals Data Dictionary.

## Appendix B. Examples of Children's Justice Goals and the Nature of Interviewer Prompts

| Justice Goal | Example of Children's Words | Example of Interviewer Prompt |
|---|---|---|
| **Acknowledgment** | | |
| Victim | "I knew I needed to tell someone because I was really scared, and I needed somebody to talk to and let it all out" | "what made you tell her" [direct] |
| | "I cried, I got upset" | "what did you do on your break?" [indirect] |
| | "I think he is trying to hurt me or something" | "say that again" [indirect] |
| | "he took advantage of me" | "okay. And what did you tell me on the video? What was that about?" indirect] |
| | "I came here to talk about me being sexually assaulted" | "tell me what you've come here to talk about today" [indirect] |
| Harm | "because I got hurt by that bully" | "why are you here today" [indirect] |
| | "it hurts" | "what does it feel like" [direct]/"yep' [indirect] |
| | "he was hurting me" | "explain that for me" [indirect] |
| | "I feel hurt and embarrass saying" | "leg, right" [indirect] |
| | "I had to go to the doctors" | "what do you mean at the top" [indirect] |
| Believed | "I didn't think anyone would believe me" | "is there a reason that you can think of why you never told anyone that could help you about this straight away" [direct] |
| | "he told me I was lying but I wasn't" | "when did you tell mum" [indirect] |
| | "she didn't believe me" | |
| | "that she wouldn't believe me" | |
| | "I told my stepdad but he thought I was mucking around" | "mmhmm" [indirect] |
| | "he thought I was lying" | "what did your stepdad do" [indirect] |
| | | "what made you decide to tell your sister at that time" [direct] |
| | "no one was listening to me" | |
| | "I told her a while ago and like that's what helps my parents believe me even more" | "what did you say to her" [indirect] |
| | "another reason why I didn't want to go back home as well because you know, mums always picking her partner over me but I didn't realise she actually didn't know" | "Yeah" [indirect] |
| | "because I was too scared that he might get really angry and well mum might not believe me" | "tell me why you didn't tell" [direct] |
| | "always when i want to tell mum something thats the truth she doesnt actually listen to me, she thinks that im lying" | "tell me why you didn't tell" [direct] |
| | "because my parents don't believe what I say except for mum" | |

| | | |
|---|---|---|
| Not blamed | "I kept it a secret because I didn't want to get in trouble" | "who was the first person you told about this" [direct] |
| | " … that anything that happened would be my fault" | "the first time when it happened, how did it make you feel?" [direct] |
| | "he made me feel bad like id done something wrong to deserve it or something like that—I know I haven't done anything wrong" | "how did that make you feel at the time" [direct] |
| | "and I thought it was my fault and that but uncle told me its never my fault" | "mmhmm" [indirect] |
| | "I didn't want to get punished" | "what made you tell your mum" [direct] |
| | "I couldn't tell my mum because I was too scared and like I didn't want her to think any worse of me" | "mmhhmm and what happened then" [indirect] |
| | "[suspect words] remember this is what you want not me" | "tell me how he did that" [indirect] |
| | "[he would tell everyone] its all my fault" | "tell me how he did that" [indirect] |
| | "he did it to me" | "any questions about..-no" [indirect] |
| | "mum said it was (suspects) fault that he did that to me" | "so I know ive obviously spoken to mum" [indirect] |
| | "[perpetrator words] you know, it's gunna be all your fault and if I get hurt, it's gunna be all your fault" | |
| | "at the time I felt like it was all my fault because I let him do it" | "yeah" [indirect] |
| **Wrongfulness** | "It's not right" | "can you tell us all about why you've come to talk to us today?" [indirect] |
| | "touching me in places he wasn't suppose to" | "tell me about that" [indirect] |
| | "he done something wrong" | "can you tell us why you've come here to talk to us today" [indirect] |
| | "he did something inappropriate with me, its called child abuse" | "Mhmm" |
| | "so then he forced me" | "tell me everything about the incident that were talking about tell me more about that" [indirect] |
| | "I just find it really wrong" | "and where does it touch you when.." [indirect] |
| | "she did something bad to me" | "what about suspect sorry darl?" [indirect] |
| **Protection** | | |
| Self | "I didn't want it to happen again" | "and what made you tell mum about this?" [direct prompt] |
| | "I told him to stop but he didn't stop" | "starting right at the beginning and tell me absolutely everything" [indirect] |
| Other | "I don't want it to happen to anyone else" | |
| | "he musnt ever do that to another person" | "mhm yeah" [indirect] |
| | "I didn't want it to happen to her" | "what was it that changed for you to tell" [direct] |
| | "I found out that he had been sexually abusing me little sister so I reported it to the police" | "tell us why you are here today" [indirect] |
| Perpetrator | "I was trying to protect (suspect)" | "mmhmm" [indirect] |
| | "no I really like him, I don't want him to get in trouble" | "yeah" [indirect] |
| Self and other | "other girls said that they were touched inappropriately but they were scared to tell the teacher" | |
| | "and we always tell each other and I said I think we should tell our mums" | "mmhmm" [indirect] |
| | "so that we can do something about it and stop it from happening" | "why do you think (cousin) told you about what happened" [direct] |

| | | |
|---|---|---|
| **Perpetrator Accountability** | "I told him again and again and again to stop but he didn't—that's why I came here" | |
| | "I was just going to wait to tell police" | "you didn't tell anybody?" [direct] |
| | "why did you do that to me?" | "so tell me everything that happened with the couple of times" [indirect] |
| | "no he didn't even say sorry" | "so tell me everything that happened with the couple of times" |
| | "I don't really forgive him" | "so tell me everything that happened with the couple of times" |
| | "how could you do that" | "how did you feel when you saw him in the morning?" [direct] |
| | "he would go to gaol" | "okay" [indirect] |
| | "I ran to the nearest shop to ask if they can ring the police for me" | "what happened after you woke up in the hotel" [indirect] |
| | "I knew she would have to call the police or have to tell the doctors" | "yeah" [indirect] |
| | "I'm here to put an end to it" | "so tell me what we are here to talk about today" [indirect] |
| | "I thought about who I could ring to tell them" | "what did you do in the bedroom" [indirect] |
| | "I've tried to remember as much information as I can so I can get this sorted out" | "was there something specific that happened a month ago that made you remember" [indirect] |
| | "I would like to charge suspect" | "mmhmm" [indirect] |
| **Punishment** | "so they can hurt that man the way he hurt me" | |
| | "well I want him to go to jail for one thing so cant go anywhere near me" | "what do you want to happen to?" [direct] |
| | "rang the police" | "mmhmm" [indirect] |
| | "I smacked him" | "what made him stop" [direct] |
| | "I hope he dies and everything" | "mmhmm" [indirect] |
| | "I just wanted to beat his head in" | "yeah" [indirect] |
| | "I want him to go to goal" | "what would you like to see happen to suspect for all the things that he's done to you and other kids" [direct] |
| | "he should get punished" | "what do you think should happen to perpetrator [direct] |
| **Rehabiliation** | "so he can help him to stop so he cant do it anymore" | "tell me a bit more about what dad said we need to stop |
| | "to help my uncle" | so can you tell us why you have come to talk to us today" [indirect] |
| | "I want him to get back to normal but he wont - so and be good again." | "hhmm is that so" [indirect] |
| | "He thinks he is going to jail but he's not he's only going to get teached to be good" | "Why did offender tell you to not tell the police [direct] |

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
