# Peer review of "Claiming Justice: An Analysis of Child Sexual Abuse Complainants’ Justice Goals Reported during Investigative Interviews"

_laws, 2022_

Round 1

Reviewer 1 Report

-Opening line, "Supporting children’s disclosures of sexual victimisation presumes that adults understand why children tell" should be reworded.

-Why not report the exact p-values in the analysis so that the reader can evaluate the level of significance?

Reviewer 2 Report

Thank you for the opportunity to review the article entitled, “Claiming justice: An analysis of child sexual assault complaints’ justice goals reported during investigative interviews".

The paper was very well-written and presented a novel examination regarding the motivations children have in reporting sexual abuse. The authors use “justice” as the interpretive lens. 

I would suggest a minor change to the title to use the word “abuse” rather than “assault” as “abuse” tends to be used more often when discussing children. 

The study followed well from previous research.

The authors had a very large sample size for this type of analysis adding to the strength of their manuscript. 

The descriptive results were appropriate as were the chi-square statistics given the categorical nature of the data.

The conclusions were well-founded. I wonder if the authors wish to note any limitations in the discussion portion of their paper and provide some future directions for further research.
